# Assessing the Feasibility of Providing a Family Skills Intervention, “Strong Families”, for Refugee Families Residing in Reception Centers in Serbia [note 1]

**DOI:** 10.3390/ijerph18094530

**Published:** 2021-04-24

**Authors:** Aala El-Khani, Karin Haar, Milos Stojanovic, Wadih Maalouf

**Affiliations:** 1Prevention, Treatment and Rehabilitation Section, Drug Prevention and Health Branch, Division of Operations, United Nations Office on Drugs and Crime (UNODC), Wagramer Strasse 5, A-1400 Vienna, Austria; Aala.elkhani@gmail.com (A.E.-K.); Karin.haar@un.org (K.H.); 2United Nations Office on Drugs and Crime, Program Office Serbia, Bulevar Zorana Djindjica 64, 11000 Belgrade, Serbia; Milos.stojanovic@un.org

**Keywords:** refugee, family skills, “strong families”, parenting, displaced population, war

## Abstract

War exposure and forced displacement threatens the wellbeing of caregivers and their children, leaving them at risk of negative outcomes, such as elevated rates of anxiety, depression and post-traumatic stress disorder. The importance of engaged, responsive and stable parenting for positive child wellbeing has been documented across diverse cultural and economic backgrounds. Despite the higher need for caregivers to be nurturing in challenging settings, they struggle to provide adequate support for their children due to lack of resources or their inability to deal with their own emotional challenges. A feasibility study was conducted of a new, open-access and light-touch family skills intervention, Strong Families (for families in humanitarian and challenged settings) on refugee families residing in Reception Centers in Serbia. Questionnaires and interviews were completed by participating caregivers and facilitators. Qualitative results indicated that the intervention was feasible to run in this humanitarian context, that caregivers viewed the intervention as culturally acceptable and complemented the quantitative results that showed promise for enhancing child behavior and family functioning tested indicators. Despite being a light intervention, Strong Families indicated improvement on child mental health, parenting practices and parent and family adjustment skills. Prioritizing family mental health and functioning as a primary need that parallels that of accessing physical medical care, sanitation and clean water must be the definitive next step in humanitarian aid.

## 1. Introduction

A staggering one in eight children are born into situations of conflict and crisis [1]. Exposure to war and forced displacement threatens the wellbeing of caregivers and their children, leaving them at risk of a number of negative outcomes such as elevated rates of anxiety, depression and post-traumatic stress disorder (PTSD) [2]. Families fleeing war experience high levels of stress, such as financial loss, family separation and emotional and behavioral changes in family members, as reactions to exposure to extreme adversities [3,4].

War exposure to conflict and crisis can influence children’s adjustment directly, by exposing them to extreme adversities, such as repeated experiences with violence, loss of caregivers or a supportive community [5,6]. Indirect effects of war exposure can sometimes be even more damaging for children, such as changes in parenting practices and their caregivers’ reactions to distressing events [7]. The importance of engaged, responsive and stable parenting for positive child mental health wellbeing has been documented extensively over the past decade [8] across diverse cultural and economic backgrounds [9,10]. Despite the higher need for caregivers to be nurturing in challenging settings, they often struggle to provide adequate support for their children due to lack of resources or their inability to deal with their own emotional struggles [11,12].

As a result of war and conflict, the number of displaced persons in the world continues to rise [13]. Refugees can experience many adverse conditions, both during flight and on reaching the next stage in search of safety, that can compound with their already intense possible war exposure. Financial struggles, living in improvised conditions or experiencing a new culture can all lead to detrimental effects on caregiver’s mental health, well-being and on their marital relationships [14]. This caregiver stress can lead to increased violence between caregivers and their children [15], and to accentuation of relationships that are far less nurturing [11].

Family skills interventions have been found to be effective in encouraging safe and supporting relationships between caregivers and children, and as such, preventing many problem behaviors including violence and other negative outcomes, such as increased substance use, risky sexual behavior and dropping out of school [16]. Family skills programs offer a combination of parenting knowledge, skill building, competency enhancement and support [17]. They aim to strengthen family protective factors such as communication, trust, problem-solving skills and conflict resolution, and strengthen the bonding and attachment between caregivers and children. There is much evidence of the effectiveness of parenting and family skills interventions in high-income and stable contexts; however, there is a lack of interventions that have been tested in low and middle-income countries and likewise in the contexts of refugee and displacement [18]. However, there are four recent exceptions of studies indicating the effectiveness of parenting and family skills interventions, in refugee and displacement contexts, in reducing child maltreatment and improving parental and child mental health in humanitarian settings [10,19,20,21].

Clearly, utilizing family skills interventions that address the needs of families affected by war, focusing on educating caregivers on better parenting their children, is an area that is increasingly being explored in the humanitarian context [22]. The underlying assumption for using parenting psychoeducation in such contexts is that change in parenting skills is a key mechanism for reducing current and preventing future child behavioral and emotional problems, for which children in such contexts may be more at risk [7,23]. Despite the increased focus on their potential usefulness and need, a recent review of existing evidence of parenting programs in low and middle income countries indicated that evidence informed interventions, that incorporate sessions tailored to conflict-affected populations, are greatly needed [6].

In response to this identified gap, the United Nations Office on Drugs and Crime (UNODC), worked with experts in the field to develop the family skills training intervention, Strong Families. The Strong Families program is a selective evidence-informed prevention intervention designed to improve parenting skills, child well-being and family mental health, amongst families with children aged between 8 and 15 years. Namely, it was tailored for families living in stressful situations including in challenged and humanitarian settings such as those experienced by refugees. The program was designed to be brief, “light” (requiring an infrastructure that is easy to mobilize and train), evidence-informed, open source (to facilitate national ownership and scale-up) and cost-effective. Initial findings of its efficacy have indicated encouraging results in Afghanistan [24].

At the time of data collection in October 2017, there were 4361 refugees in Serbia registered with the United Nations High Commissioner for Refugees (UNHCR) Office in Serbia, of these 82% were from Afghanistan [25]. Most refugees (93%) were accommodated in Reception Centers facilitated by the Government of Republic of Serbia while they waited to proceed further to Western Europe. Serbia was perceived as a transit country by the majority of refugees and had low cases of asylum applications [26].

The aim of this study was to evaluate the feasibility of delivery and any potential impact of Strong Families with refugees. Specifically, the study examined the feasibility of recruiting local non-specialist research staff and intervention facilitators in challenged settings; and recruitment of caregivers as well as their engagement and retention in the intervention and study. The second aim was to assess potential benefits of the intervention for families, in improving family functioning, children’s psychological wellbeing and its cultural appropriateness.

## 2. Materials and Methods

A mixed method approach using quantitative and qualitative measures was used. Quantitative measurements evaluated potential effectiveness including a prospective collection of outcome data assessing changes in children’s behavior, parenting skills and family adjustment in caregivers. In addition, qualitative interviews facilitated understanding of the caregiver’s experience in taking part in the intervention and cultural acceptability.

### 2.1. Participants and Procedures

Participants were selected based on being a primary caregiver to a child between the age of 8–15 years. Sampling was opportunistic, using a universal approach, in which facilitators recruited refugee families from Reception Centers, not targeting specifically those with any risk. The study was advertised in the three Reception Centers in Southern Serbia as designated by the Ministry of Labour, Employment, Veteran and Social Policy of the Republic of Serbia (MLEVSP). All refugee families in these centers were invited and the study adopted a self-referral approach. Inclusion criteria included being a refugee, willing to take part in the intervention and being in the Reception Center for the duration of the study and measurement meetings. Families in which the caregiver resided separately from the child were excluded.

Overall, 26 families were invited to participate in the study, 12 from Vranje reception Center, 11 from Preševo reception Center and 3 from Bujanovac reception Center. All families agreed to participate, met the inclusion criteria and participated in the program. One family was excluded after leaving the camp after taking part in the pre-intervention assessment; therefore, overall 25 families participated in the study. These were 20 mothers and 5 fathers to the participating children. All 25 were from Afghanistan, agreed to take part in the study and completed the program and all data collection. The three refugee Reception Centers designated by MLEVSP in October 2017 were in the towns of Vranje, Preševo and Bujanovac. The MLEVSP selected these Reception Centers to meet the study inclusion criteria and where at least two rooms were available to run the intervention. At the time of the study the Reception Center in Presevo had an occupancy of 210 refugees, in Vranje, 210 and Bujanovac 73. The average duration of stay in these Reception Centers was between 6 months to 1 year. Refugees were stranded in Serbia due to Hungarian migration procedures, often leaving refugees unaware when their journey will continue.

Despite the intention not to restrict the intervention to a specific nationality, coincidentally the reception centers selected housed over 80% Afghan nationals. Some other nationalities were also present in these centers; however, they were mostly male or not living as family (or with accompanied children) and as such did not meet the eligibility criteria.

Caregiver self-referral information sheets on Strong Families were distributed to all caregivers that had children aged 8–15 inviting them to an information session. The information sheet was also read to caregivers to ensure those who could not read would not be excluded. Following the information session, caregivers were asked to approach the Reception Center manager within the next 3 days if they wanted to participate. All attendees of the information session accepted to attend the baseline measurement session the following week. During this session, written informed consent was obtained prior to data collection, and as all children were under the age of 16 years, parents consented for their children. The Strong Families program was delivered to families in groups of not more than 7 families. The restricted number was to ensure manageability of the session given the need to run consecutive translation of sessions content as delivered by the Serbian speaking facilitators. Only one parent or caregiver (the primary caregiver) was invited to attend with a maximum of two children, one of them being in the age range of the intervention.

The only criteria for facilitators nomination was that their current role required them to have direct access to caregivers and their children in the three study settings. Accordingly, twenty facilitators, from local civil society organizations (CSOs) working in the Reception Centers, were nominated by the MLEVSP. These CSO nominees had daily contact with refugees in these Reception Centers and were familiar with the families’ needs and challenges. While six facilitators had experience with family skills programs, of which five were social workers, the majority did not have a specific experienced background. This mix of backgrounds was important as the program is designed for lay facilitators with no particular expertise. Facilitators took part in a 3-day Strong Families training in Serbia, delivered by the developers of the program who are experienced international trainers. Selected interpreters and research assistants also took part in the training to acquire information on program material, terminology and its execution.

### 2.2. Confidentiality and Ethical Considerations

This study has been reviewed and approved by the UNODC Drug Prevention and Health Branch in the Headquarters (HQ) office of Vienna and the national UNODC project office in Serbia, as well as the MLEVSP and the selected CSO that supported the program. From the UNODC perspective, the objective was to further enrich the experience in feasibility, practicality and benefit of Strong Families that assessed and documented in Afghanistan [24] in a refugee setting. All procedures performed in studies involving human participants were in accordance with the ethical standards of the institutional and/or national research committee and with the 1964 Helsinki declaration and its later amendments or comparable ethical standards.

Confidentiality of participants was ensured in accordance with the Data Protection Act 1998. Participants were assigned a unique identification number to ensure matching of all questionnaires and interviews. All data collected as part of the trial were treated as confidential and only viewed by members of the trial team; anonymized data were used wherever possible. Participants were allowed to drop out of the program any time without further justification/explanation.

### 2.3. Data Collection

Data on demographics, emotional and behavioral difficulties and parental skills and family adjustment measures were collected from caregivers through self-administered questionnaires. Translation into Dari had been assured and caregivers needing help were supported by the facilitators and translators. Two research assistants, trained by the program developers, supported the data collection.

At baseline, a Family Background Questionnaire was used to collect demographic characteristics. Further, two different paper-based questionnaires, Strengths and Difficulties Questionnaire (SDQ) and the Parenting and Family Adjustment Scales (PAFAS), were used as repeated measures. These were filled in at baseline (Pre-test; t1), two weeks after the intervention (Post-test; t2) and six weeks after completion (Follow-up; t3).

The SDQ [27] is a widely used tool completed by caregivers to screen children for emotional and behavioral difficulties over the last six months. It is available in over 40 different languages and is frequently used for research purposes to examine children’s mental wellbeing. The advantages of the SDQ were its compact format (relative to the previously long-established and highly respected Rutter behavioral screening questionnaires) in covering the strengths as well as difficulties in inattention, peer relationships, and prosocial behavior [27]. It has shown good psychometric properties and has been used in Afghanistan previously. The Dari versions had demonstrated good internal reliability (Cronbach α = 0.66 for caregiver-rated SDQ total difficulty scores, *n* = 364) and test-retest reliability (Spearman Brown r = 0.57, *p* = 0.009, *n* = 20). [15]. Twenty-five items, rated on a 3-point Likert scale ranging from 0 (“Not True”) to 2 (“Certainly True”), form five subscales, each with five items, including items such as emotional symptoms (e.g., “Often unhappy”) and conduct problems (e.g., Often fights with other children). Total scores for subscales range from 0 to 10 and the overall total score (sum of the subscales except for prosocial behavior) ranges from 0 to 40, with higher scores indicating higher levels of difficulties. The 4-banded categorization was used to classify continuous measures into “close to average”, “slightly raised”, “high” and “very high” risk.

The PAFAS [28], is a 30-item, questionnaire that measures parenting practices, risk and protective factors, such as parental emotional adjustment and quality of family. It consists of two scales: (i) Parenting, that comprises four subscales and (ii) Family Adjustment, that comprises three subscales. The subscales show good internal consistency (ranging from 0.70 to 0.87). The parenting subscales “parental consistency”, “coercive parenting” and “family–child relationship” range from 0 to 15 points and “positive encouragement” from 0 to 9 points, higher scores indicating worse outcomes. The family adjustment sub-scores “Parental Adjustment”, “Family relationships” and “Parental teamwork” range from 0 to 15, 12 and 9 points, respectively. In addition, PAFAS has been validated in other cultures, including Panama [29] and families living in political conflict in the West bank [30].

During the second data collection meeting (t2), caregivers were asked if they would like to take part in interviews. They were given written and verbal information about this interview process, including its length and the questions to be asked. Those that were interested were given a card with the time and location of the interview. All interviews took place within two days of caregivers attending the second data collection meeting.

### 2.4. Data Analysis

Quantitative outcome data were entered into an Excel spreadsheet and analyzed using SPSS (version 26; IBM, Armonk, NY, USA). A Shapiro–Wilk’s test and a visual inspection of the histograms, Q-Q plots and box plots showed that data were approximately normally distributed. Data completeness was checked, as well as plausibility testing performed. No missing data was found. Continuous variables are presented as mean and standard deviation (SD), whereas for descriptive analyses of categorical data, frequencies and proportions were calculated. To compare demographic data, a 2-sample t-test was used for continuous variables and a chi-square test for categorical data. Reliability analyses were not attempted as our sample size was considered being too small, with more items in the scale than the actual sample size [31]. One-way repeated measures ANOVAs with post hoc tests using Bonferroni adjustments were conducted to compare outcomes on the continuous SDQ and PAFAS scores. In case Mauchly’s Test of Sphericity indicated that the assumption of sphericity had been violated, a Huynh–Feldt correction was used. For comparison of the SDQ scores between boys and girls, we first tested a potential group-interaction effect through a two-way mixed ANOVA with within and between subjects’ factors. We further tested the effects of the respective outcome variable for girls and boys separately through a repeated measures ANOVA [32]. Stratified analysis was performed separating participants with high (17+) or very high (20–40 points) scores at baseline on the total difficulty scale of the SDQ from those with scores below, in order show potential effects in those with most difficulties at start point. Similarly, caregivers with PAFAS scores above the 70th percentile in each subcategory were compared to those with lower scores to compare the effects on families with high problems at baseline to those with less difficulties. Significance level was set at 0.05.

Qualitative data consisted of interviews that were conducted with caregivers. These were analyzed using a mixed approach to thematic analysis (TA) [33]. TA was chosen because of its ability to directly represent the descriptions of respondents’ viewpoints, experiences, beliefs and perceptions. TA has often been used in the analysis of the experiences of displaced refugees e.g., [3]. An essentialist method was used, aiming to report experiences, meanings and the reality of participants [34]. Initial inductive coding was carried out with the aim of seeking a descriptive account of the data, rather than an interpretation and explanation of the discourse, allowing the themes to evolve from the data set rather than being theoretically driven. The team reflected on identified codes to combine and rename these where appropriate. The team developed a revised code set that included the new and combined codes. Links between, and within, themes were also examined. The research team reviewed the emerging themes and came to an agreement on final themes.

### 2.5. Program Intervention

The Strong Families program [35,36] is a three sessions (5-h of invested time by the families in total) group intervention for primary caregivers and their children lasting over 3 weeks (one session per week). In the first week, a group of up to 10–12 caregivers meet for the 1-h caregiver pre-session. In the second week and third, the same 10–12 caregivers meet again in one room, and their children meet in parallel in another room for the child sessions. After these 1-h parallel sessions, all caregivers and children immediately meet in one room for another1-h joint family session.

During the first week’s session (caregiver pre-session), caregivers explore the strengths and skills they already have and also their challenges and develops ways to better deal with stress. In week two, caregivers learn tools to show their children they care about them while also having and enforcing limits. while the children learn how to deal with stress. During the family session they come together to practice positive communication and practice stress relief techniques together. In the third week, caregivers learn how to encourage good behavior and discourage misbehavior, while children explore and take part in activities about rules and responsibilities and think about future goals, in addition to the important roles their caregivers play in their lives. In the final family session, caregivers and children learn about family values and practice sharing appreciation to each other.

The version of Strong Families used in this study was previously culturally adapted, translated and reviewed in Afghanistan and successfully piloted on Afghan families in three Afghan cities [24]. The program was translated to Serbian for the purpose of training and implementation in Serbia. More information on the content of the sessions of the Strong Families program can be found on the UNODC website [36]. 

## 3. Results

### 3.1. Quantitative Data Results

#### 3.1.1. Demographic Results

All 25 participating families reported having fled Afghanistan in response to armed conflict in their home country, and had been living in Serbia for one year on average (±0.14). Overall, there was no difference between gender of caregivers or children regarding the demographic data. Mean age of caregivers was 33.4 years (±5.99), 88% were married, 56% of caregivers and 48% of partners had primary school education or less. Forty percent of primary caregivers were unemployed, as were 32% of their partners. On average, caregivers had 3.3 children (±1.7). Families attended with one of their children in the age group as defined in the inclusion criteria. The mean age of these 9 girls and 16 boys did not differ significantly (10.5 years (±2.38)). Mean age of children enrolled in the program was higher than the ones of their non-attending siblings (Appendix A) In 80% their mother took part with them in the program, in 20% the father. Mothers participated with 9 daughters and 11 sons, 45% and 55%, respectively; however, fathers only participated with sons (Appendix A; not significant).

#### 3.1.2. SDQ Results

Overall, a significant reduction in the “total difficulty scale” scores could be seen before and after the program (t1–t2; *p* = 0.004), this reduced score was maintained at t3 (*p* = 0.002). This overall observation was replicated in both, boys, and girls (*p* = 0.011 and *p* = 0.017overall). Aside the reduction in average score, it was worth noting that the highest maximum scores were consistently recorded at baseline decreasing at t2 and t3 consistently for both genders (Table 1).

On the SDQ subscales, emotional problem scores were statistically significantly decreased after the intervention (t2; *p* = 0.001)) as well as overall (*p* < 0.001) in both genders with a sharper noted decrease in scores in girls (*p* = 0.004 overall). Conduct problem subscale was also noted to drop in scores from t1 to t2, reaching a significantly lower score at t3 (*p* = 0.001). This reduction in scores was also noted across genders (*p* = 0.014 in boys and *p* = 0.034 in girls overall). The hyperactivity subscale score overall was reduced from t1 to t2 reaching the lowest level at t3 (*p* = 0.012 overall). Upon inspecting these results by gender, this seems to be driven mostly by change in scores in boys (*p* = 0.02 overall) and less so in girls where the score did not significantly vary over time, however boys starting off at a much higher level compared to girls. There was, however, no statistically significant interaction between gender and time on hyperactivity scores, as shown in Table 1. Peer problems and prosocial behavior scores did not significantly change after the intervention; however, measurements at baseline were already close to average, apart from peer-problems in boys, which was slightly raised (Table 1).

Nine children, 8 boys and 1 girl, had high (17–19 points) or very high (20–40 points) total difficulty scores at baseline (scores of clinical importance). Overall, their total difficulty scores decreased significantly at t2 (*p* = 0.021; with the average score below the clinical cut-off) and continued to do so at t3 (*p* < 0.001 overall). All SDQ sub-scores, apart from the prosocial behaviour, improved significantly from t1 to t3 (Overall *p_Emotional problem scale_* = 0.006; *p_Conduct problem scale_* = 0.007; *p_Hyperactivity scale_* = 0.005; *p_Peerproblem scale_* = 0.013), with the lowest sores recorded at t3, continuing the improving trend also between t2 and t3 (Table 1).

#### 3.1.3. PAFAS Results

A significant reduction in scores was noted on all parenting subscales (except for parental teamwork) and on the family adjustment subscale, parental adjustment, between t1 and t2 in families who scored over the 70th percentile on each PAFAS subscale at baseline (Figure 1). Within this subgroup of families, significant changes over time were found on the all the following subscales: Parental Consistency (F(2,16) = 6422, *p* = 0.009, partial η^2^ = 0.45), Coercive Parenting (F(2,14) = 26.275, *p* < 0.001, partial η^2^ = 0.79), Positive Encouragement (F(2,20) = 7.679, *p* = 0.003, partial η^2^ = 0.43), Parent–child Relationship (F(1.200, 10.798) = 9.387, *p* = 0.009, partial η^2^ = 0.51) Parental Adjustment (F(2,14) = 15.494, *p* < 0.001, partial η^2^ = 0.69) and Family Relationships (F(2,14) = 4.200, *p* = 0.037, partial η^2^ = 0.38). Scores reached the lowest values at t3, being statistically significantly lower than at t1 in post-hoc tests within the Coercive Parenting, Parent–child Relationship and Parental Adjustment subscales. Families scoring above the 70th percentile on baseline (t1) on parental teamwork while consistently lowering the average score from t1 to t2 and then to t3 did not register a statistically significant change overall.

Families with scores below the 70th percentile at baseline did not show significant changes in scores in any of the dimensions. Overall, scores for positive encouragement, parent–child relationship, family relationship and parental teamwork were at a decreased level already at starting point, leaving little room for improvement. Despite this, all scores moved in the right direction in all subcategories.

### 3.2. Qualitative Data Results

Overall, eight participating caregivers and six facilitators took part in interviews. Table 2 presents the gender and age of participants. Three main themes emerged: changes in parenting practices, improved communication, and engagement and satisfaction. Each is described and illustrated with quotes provided (identified according to Caregiver (C), or facilitator (F)). Figure 2 illustrates the thematic map of the results of the qualitative analysis.


**Theme 1: Changes in Parenting Practices**


Parents talked extensively about how they perceived improvements in how they parented their children. One mother described how she felt the intervention helped her by gaining increased insight about her child and increased empathy, therefore helping her manage her expectations, and this is what led to her changing how she parented her child.


*“I was always angry at what they were doing, thinking they did something serious. But now I see it differently, so I act differently”.*
 *(C3)*


**1.1. Reduction in Physical Chastisement**


The discussion often revealed caregivers were using high levels of physical chastisement. The majority of the caregivers (seven) stated that they used much less punishment with their children since taking part in the intervention. One caregiver said:


*“To tell the truth I used to beat them every day, now I don’t do that anymore, I try to verbalize it, not start hitting them immediately. That’s different. I also get a better effect from them this way”.*
 *(C2)*

Another added,


*“It helped me and other people, because sometimes people would want to hit their kids, they used to hit them before, but this program has helped us behave in a different way. It’s hard in this camp and then we get angry, I would slap the kids a little before, but now it’s better”.*
 *(C8)*

On being asked what caregivers felt were the reason they were hitting their children less, one single mother described how the sessions she took part in directed her to manage her feelings and be more in control of how she reacted. She said:


*“I can now control myself better. When I am angry I tell myself ‘I must not behave this way’, and then I behave much better. I learnt to control myself…and actually they listen more, they listen well”.*
 *(C4)*

Similarly, another mother attributed the self-control she learnt as key to reducing physical punishment:


*“I learnt about the kids for example to help [them] more, listen to them more, hear them out, hear what they say, control myself more, have more self-control. This feels very good and makes me happy and them happy too”.*
 *(C8)*


**1.2. Prioritizing Caring for Children**


Several caregivers described how the challenges of conflict and displacement had affected their parenting, often forcing them to just focus on keeping their families alive rather than parent their children. One mother disclosed how she had been struggling with these feelings, and how the intervention helped improve this, she said:


*“We had a lot of trouble along the way, so that we forgot we were mothers, then we lost that feeling, the feeling of being a mother. Now I remember again. I know how I should act with my children. As a mother, to be a real mother again”.*
 *(C7)*

One facilitator further highlighted this:


*“The parents are really busy thinking about what they will do next [their future], and then maybe at some point the children end up in the background. I think that in this way [taking part in the intervention], the children were put to the foreground a bit and I think the children liked it. I think it gave them importance”.*
 *(F4)*

One mother described how in their country of origin, how to care for children was not a topic they thought much about or was given focus due to other challenges:


*“In our country there has always been war, we never had the opportunity to think about things like this. No one even asked us about our children or told us how to act with them”.*
 *(C2)*

Caregivers spoke greatly about their increased confidence or understanding of their children’s needs and how they should care for them. Often this was emotional for them to describe and gave them a sense of great accomplishment that they now were giving their children what they needed. One father said:


*“Before I didn’t know how to look after them [children] right. Either I would show too much, love or be too strict. Now I have learnt how to explain things. Several days ago, one of my children did not want to go to bed, but I said, ‘you need to be in bed on time’ and explained why. I knew what to do”.*
 *(C5)*

Facilitators also noted caregiver’s improved ability to become aware of their children needs and respond appropriately. One said:


*“They have only now become aware of some of their children’s feelings and that the children feel this situation and everything that’s happening to them. We had some emotional reactions in some. I really think it was successful and it was meaningful”.*
 *(F3)*


**Theme 2: Improved Communication**


A central idea focused on commitments to better communication. Good communication, both within caregiver couples and between caregivers and their children was emphasized as a major positive outcome of taking part in the intervention.


**2.1. Parent–Child Communication**


One facilitator who had been working with many of the participating families for up to a year noted a positive change in family communication:


*“Communication in the family is not at the level it should be. Of course, this is expected in such life circumstances, but the communication is really bad. Children didn’t listen to their caregivers enough, and don’t even have anyone to talk to about their problems and everything, and I wasn’t much surprised by that picture. Now they share and both talk to each other”.*
 *(F2)*

All caregivers said they wanted their children to listen and respect them. They identified how the changes in the way they were now communicating with their children, after taking part in the intervention, was facilitating this for them. One father noted this better communication led to improvements in their child’s compliance and behavior;


*“We learnt to talk with this tone, a different tone that we practiced. This is really good for the kids. Not to talk so angrily to the kids but to use another tone. Now it’s better, much better really. I talk more to them and in a kinder way and then they in return behave better”.*
 *(C1)*

Another caregiver added:


*“I was always angry with the kid. In fact, I didn’t even know why the kid was fighting. I learnt that I need to talk with the kid. Now I talk and I know how to react. I know what’s going on, so I react appropriately”.*
 *(C3)*


**2.2. Intercouple Communication**


Participating caregivers who were present in the Reception Centers with a partner, described how after taking part in the intervention, caregivers were now making time and prioritizing communicating with their partners about their children. One mother said:


*“We would hardly talk about how the kids should be, now we talk every couple of days for about half an hour, perhaps an hour when there’s a problem. About how we should treat them [children]. This was certainly not how it was before the program”.*
 *(C2)*

Caregivers also highlighted a reduction in conflict between caregivers within families on completion of the intervention. One mother said:


*“Before, both of us [husband and wife] would argue a lot. We would fight about our son and because he [husband] was always angry at him [son]. But we learnt that this is not how it should be, no one is benefitting. We all act differently and talk better now. He [son] acts differently and so do we, and so then there is much less fighting”.*
 *(C3)*

Participants spoke extensively about marital challenges and conflicts in their marriage, often describing a breakdown in communication. One caregiver described how taking part in Strong Families gave her husband and herself an opening to opportunities to talk more with each other and how this has expanded to other topics of communication between them. She says:


*“We began talking to each other, we had a shared topic to discuss. I was the one who took part, so I told him what I had learnt and I also told my other kids. He was receptive and took it in. We soon started talking about other things again, sometimes good memories and sometimes hard things”.*
 *(C1)*


**Theme 3: Engagement and Satisfaction**



**3.1. Cultural Acceptability**


When cultural acceptability was explored, all caregivers and facilitators responded that the Strong Families intervention was culturally appropriate and gave no indication of challenges in any beliefs or norms. One mother indicted how closely the intervention fitted her value system:


*“I grew up in a Muslim family and real Islam is like that; the family is paramount. The relationship between family members should be good, just like we learnt in this program. What we learnt [in the intervention] is all in our religion; to respect each other, our families as well and respect the elderly”.*
 *(C4)*


**3.2. Identifying Other Needs**


Facilitators reported that even though they were quite familiar with the participating families, running the sessions helped them identify children that had other problems that required additional support. One explained:


*“We identified two children, that we would later on start working with further, during the sessions we noticed signs of stress like bed-wetting which is very alarming and must be addressed. This wasn’t explored by the doctors previously. It was effective [the intervention] in the sense of identifying the signs for the identification of children’s needs, so that some form of individual work may be later on designed”. *
 *(F1)*

Another facilitator suggested that caregivers became able to distinguish between normal challenges and those that could indicate children needed further intervention and support unrelated to caregiving.


*“They [participating caregivers] and us [facilitators] learnt that some symptoms, which we may think are harmless, are actually very worrying and not so harmless. We learnt to be aware of signs of stress and challenges that are not normal reactions to the situation of these families. Caregivers learnt that spending more time with their children is what will help you identify such things”.*
 *(F1)*


**3.3. Usability**


Caregivers were able to recall different activities and described how they had begun to integrate them into their daily practices, shaping and sharpening their skills as caregivers.


*“The deep breathing exercise is very good. I go to the terrace and there I breathe deeply and it feels really nice”.*
 *(C5)*


*“I have learnt not to dwell on stress a lot. I now understand it does not make much sense to dwell on it and it makes me unable to be a good father when I do. I have learnt tapping and breathing, and these are very useful in my life now generally”.*
 *(C6)*

All caregivers also commented that children enjoyed the intervention and that this further motivated them to continue participating in the weekly sessions.


*“Our children were thrilled with the program and kept asking when the next sessions were”.*
 *(C2)*

Three facilitators suggested that the program reach should be expanded and that everyone in the centers could benefit. One facilitator said:


*“Strong Families should be made accessible to everyone in all Reception Centers. Not only where we are. It’s well suited for other nationalities, and for us too [Serbian facilitators]. *
 *(F1)*

Participants described that the program was an easy fit and well suited for the Reception Centers as they were bored and did not have much to do to fill their time. The program provided much needed engaging activities and stimulation and also an opportunity to connect with others. One father in particular described reduced feelings of loneliness since taking part:


*“We are in a closed camp and we have got nowhere to go, and it was really good that we participated, had a chance to talk and get to know new people, feeling like we are not alone. That was good, it’s really boring in the camp and parents also like to do something, something new”.*
 *(C4)*

Another facilitator demonstrated this by adding:


*“Every day it’s the same, just eat, sleep. It’s not right at all. The program motivated them, it was dynamic and interesting. The topic well covered, to really talk and to be useful for them”.*
 *(F6)*


**3.4. Suggested Modifications**


All caregivers expressed an interest in continuing to learn family skills and more involvement. Several said they *“don’t want it to end”,* and both caregivers and facilitators believed expanding the sessions further than three weeks would be beneficial.

The majority of the caregivers responded to questions about specific sessions or activities by saying things such as *“there is nothing to change”,* though two suggested they would like to see a session on dealing with problems in marital relations. One modification that several caregivers, and all facilitators suggested, was that the program should be attended by both caregivers (when present). One Facilitator said:


*“It’s best for both parents to be there, so both can get the experience and learn. If there is a parent who is ignorant to the needs of the child and needs to understand the child more and to build this relationship with the child, being informed second hand from their partner won’t be enough”.*
 *(F3)*

## 4. Discussion

The results of this study suggest that the Strong Families program is practical to implement with refugee families displaced by conflict. Recruitment of facilitators and their subsequent training was also feasible, in addition to recruitment of participants, and maintaining attendance with high retention in the program. Despite being a light intervention, Strong Families indicated improvement on the child mental health indicators, as well as the parenting practices and parent and family adjustment skills. These results are consistent with a pilot study of the same intervention with 72 Afghan families residing in Afghanistan [24]. While the study by Haar and colleagues aimed to assess the feasibility of implementation in a setting of stress, this study intended to assess the potential replicability of the findings in a further challenging context (displacement and refugee transition), as well as differential implementation circumstances.

This study indicated that caregivers experienced Strong Families as acceptable and reflected potential for positively changing parenting behavior, child behavior and family functioning. Such conclusions need to be corroborated by additional research to document efficacy and impact. The acceptability and positive changes in scale scores are interconnected as both have implications for the intervention’s usability. Caregivers deeming the intervention as culturally acceptable and meeting their current needs will be far more likely to take part and engage and try the activities presented. Similarly, the presented efficacy results indicate that caregivers were engaged in the intervention and were inclined to try the caregiver methods taught. Quantitative results were further echoed by the qualitative analysis. Comments about the caregiver’s ability to make changes in their parenting behaviors and their experiences within their family unit indicate that the sessions were taught well enough to have a learnt effect.

Even with the modest sample size, some statistically significant changes were observed. The significant improvements in Total Difficulty score of the SDQ are promising, and the fact they were maintained even on follow up is an exciting outcome. What was most interesting was that the most challenged children with the highest levels of difficulty, assessed through an SDQ score of 17 points or more, indicating potentially clinically apparent disorders, decreased significantly right after the intervention and even more after the follow-up period. When we explored this further, we found improvement in all subscales of the SDQ (except prosocial behavior) between post-test and follow up. This result indicated that Strong Families seemed to benefit children with higher levels of difficulty the most.

Our results suggest that the intervention led to parents’ subjective report on reductions in child behavioral and emotional difficulties, supporting previous research that even very light touch interventions can bring about change to families in low resource challenging settings [24,30,37]. These outcomes correspond to research demonstrating the positive relationship between improving parenting and resultant positive outcomes for children, such as their improved emotional and behavioral regulation and socialization in conflict [8,38]. Our qualitative strand suggests that these overall improvements may be driven by the noted improvements in communication between caregivers and their children and reduction in harsh parenting practices. Separating individual components of change is a valuable task; however, the results of the potential changes in parenting are complex. For example, caregivers talked greatly about reducing physical punishment with their children, but the reasons are unclear. For some it may be due to the information learnt in the intervention about the potentially damaging effects of doing so, but for others it may be that by using the more positive techniques taught such as positive communication, there was less need for physical discipline.

Interestingly we found that in several cases, changes in results were largest at the last measure point, 6 weeks post-intervention. This parallels previous research of a randomized controlled trial (RCT) of a short parenting program in Panama (a low-resource country with high levels of violence), in which the effects of the intervention on parental reports of behavioral difficulties were moderate at post-intervention but largest at the final follow up point [37]. One reason for the increased effect with time may be that the psychological processes enhanced by the program, such as an increase in parental self-confidence and problem-solving, may prompt behavioral changes in both the parent and the child and thus with time and further practice, these bring about more improvements. Explanations behind these compounding effects with time merit further exploration.

A notable outcome of caregiver participation in Strong Families is the seeming improvement registered on caregiver self-efficacy, as illustrated by the qualitative data. Participants discussed often feeling unable to parent their children as they were overwhelmed with their experiences of displacement and focusing on essential needs. They described feelings of confidence and control as they were reminded of the importance of their role in caring for their children though the intervention sessions. These caregiver changes parallel a study in which Syrian refugees stated that interactions and support from professionals regarding caring for their children in refugee camps increased their self-efficacy and helped them re-focus on meeting their children’s emotional needs [11]. Research indicates a reduction in parental self-efficacy might lead to elevated levels of harshness [39], and this was in line with our current study as we noticed both an improvement in self-efficacy and a reduction in caregiver use of physical chastisement on children.

With the PAFAS measure we aimed to assess the improvement of parent and family functioning skills, that are known to be protective or risk factors for child emotional and behavioral problems. We found that caregivers who scored over the 70th percentile at baseline improved significantly throughout all dimensions. These improved factors, such as parenting adjustment and quality of family relationships, are known to be related to child outcomes and they are common targets of evidence-based parenting programs and are expected to change as a result of a successful parenting intervention [40]. Again, this parallels a study conducted in Afghanistan using Strong Families [24] that found similar results, thus allowing us to also conclude that the Strong Families program seems to positively target these areas of promoting a child’s positive and prosocial behavior, reciprocal warmth and understanding for each other, support to parents to deal with stress and anxiety, setting rules and targets to support a conflict-free family environment and for caregivers to support each other [28].

That caregivers found the intervention culturally appropriate as well as context relevant was a very promising finding, as a recent review suggests that current parenting program models do not address the complex factors influencing caregiver and child experiences related to their lived experiences [18]. Facilitators too described feelings of hoping to be able to go on and use the intervention with all families they had access too, further supporting the case for the interventions content being relatable and sensitive to the needs of families displaced by conflict. The study setting is also noteworthy, as it was unclear what exactly was the participants’ motivation for voluntarily taking part in the intervention. While several caregivers described feeling bored and not having much else to do to on a daily basis, others mentioned that the advertised intervention appealed to them as they felt they needed help in caring for their children. It may be fair to draw parallels with studies of caregivers that are in prisons, and offered voluntary participation in parenting intervention and have shown interest in attending a parenting class [41]. Whatever the initial motivating factor for participation may be, the 100% engagement rate as well as the interviews with caregivers indicate that throughout the three weeks of the intervention caregivers enjoyed the sessions and were motivated to continue to learn further.

Our findings do not come without limitations. The sample size and the unique characteristics of the study site (Reception Centers for refugees in transit) precludes generalizability of findings to refugees or those displaced in other settings. Secondly, our sample consisted of being entirely Afghan Dari speaking refugees. While the underlying constructs of parental engagement, communication, warmth and harshness hold universal value [42], the specific expression of these constructs can often be culturally specific [7]. Thus, our findings could not be expanded beyond the experiences of Afghan refugees. Thirdly, from each family unit only one caregiver was invited to participate with one child even when two caregivers were present, this was due to practical restraints in space and room availability and required interpretation services from Serbian to Dari language. We therefore cannot assess the impact of both caregivers taking part. Our next phase in this project is to conduct a pilot RCT of the Strong Families intervention with refugee families through a thorough outcome and impact evaluation, inviting two caregivers (when applicable) and exploring the impact of this training, also allowing to control the effects against a non-intervention group. While this study cannot be labelled as an effectiveness study, given that it is an extension and replication of the previous pilot study implemented in Afghanistan [24], the current changes detected in the current study scales tends to suggest a positive potential effectiveness of the Strong Family program that still needs to be replicated in a more elaborate study. Of added value and interest to us, is to explore the long-term follow up of participants up to one year from intervention implementation, this would be especially interesting with refugee families who may be by then resettled in new contexts, and outside the closed community this current study took place.

Despite these limitations, several strengths give us confidence in our findings. First, our mixed methods approach allowed us to integrate findings from quantitative and qualitative findings. Doing so we were able to better understand the experiences of caregivers taking part in Strong Families. Secondly, an important consequence of families participating in the intervention is the identification of children that had physical and emotional needs that required further intervention, such as a child that was experiencing severe bed wetting that had been unnoticed by center staff until the intervention implementation, and was transferred to the support of a specialist medic. The setting was well connected to other local services, thus referrals for external support was available when needed. Even facilitators that had worked with some of the participating families for up to a year were surprised that these needs had not been picked up by them or other professionals providing their services earlier. In humanitarian contexts of scarcity of physical and mental health services, the provision of universal psycho-social support is often a key means of identifying those that need more extensive or directed support. Finally, we gladly found the intervention affected girls and boys alike, this is important considering the value the United Nations is placing on gender sensitive responses.

## 5. Conclusions

The increasing number of displaced and refugee families globally indicates that there is a high need to address their collective and individual mental wellbeing and functioning. Prioritizing family mental health and functioning as a primary need that parallels that of accessing physical medical care, sanitation and clean water must be the definitive next step in humanitarian aid. Few family skills interventions implemented in conflict-affected settings have curricula designed to meet the needs of children and caregivers in this context. Strong Families might be able to ameliorate some of the effects of war and displacement by targeting family skills such as communication between all family members, warmth, reducing harshness and increasing engagement between caregivers and children. These have the potential to improve children’s well-being and to strengthen resilience. This should be given a higher priority by organizations providing support and services as well as policy makers.

## Figures and Tables

**Figure 1 ijerph-18-04530-f001:**
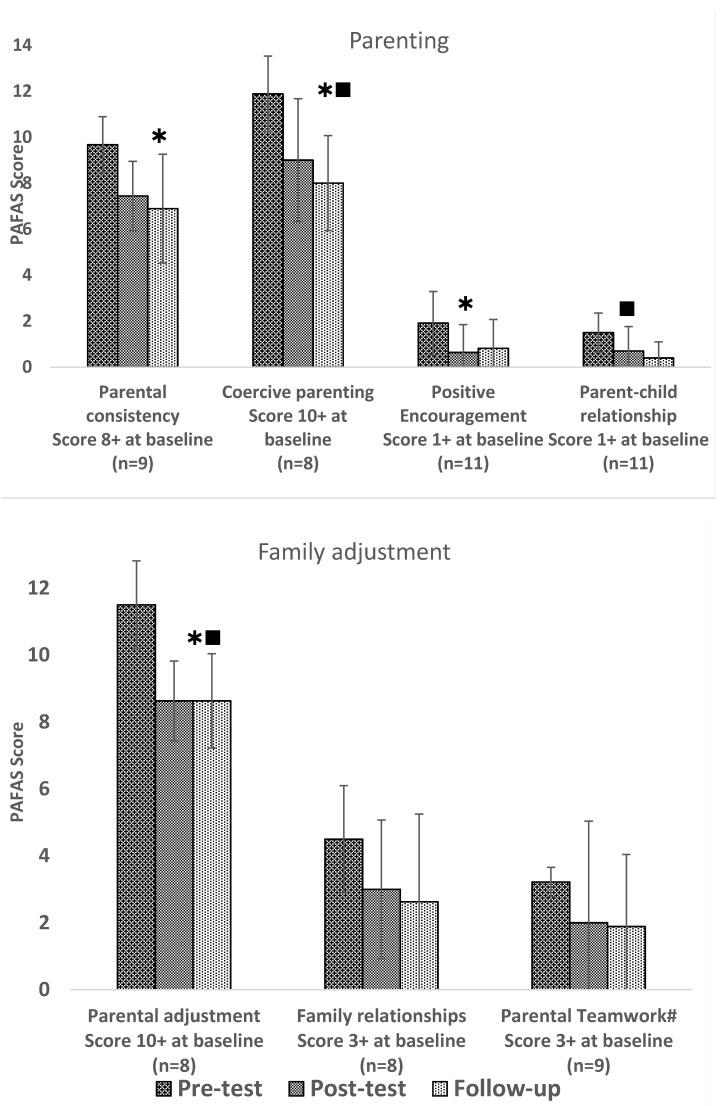
Number of families (*n*) and mean scores (and SD) of families scoring above the 70th percentile on each of the subscales of the Parenting and Family Adjustment Scale (PAFAS) at baseline (Pre-test; t1). Follow up two weeks (Post-test; t2) and six weeks (Follow-up; t3) after the intervention. ^#^ Scores on parental teamwork only included in analysis if in a relationship (1 divorced/separated and 2 widows excluded); If statistically significant in one-way repeated measures ANOVA, post-hoc-test results displayed as 
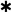
 significant difference between t1 and t2, 
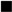
 significant difference between t1 and t3.

**Figure 2 ijerph-18-04530-f002:**
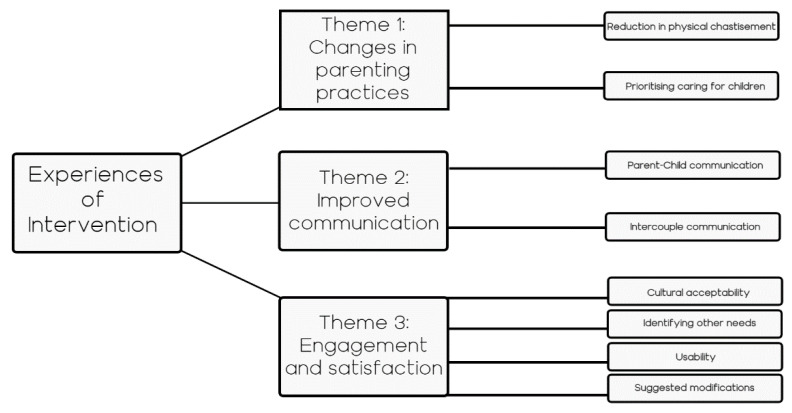
Thematic data map representing experiences with the Strong Families intervention.

**Table 1 ijerph-18-04530-t001:** Differences in scores of the Strengths and Difficulties Questionnaires before the intervention (Pre-test; t1), two weeks after the intervention (Post-test; t2) and six weeks after completion (Follow-up; t3) between girls and boys and those with very high (20–40 points) or high (17–19 points) scores on the total difficulty scale.

Gender-Based Analysis		Pre-Test Mean (SD) {Min–Max}	Post-Test Mean (SD) {Min–Max}	Follow-Up Mean (SD) {Min–Max}	Two-Way Mixed ANOVA F(df_time_, df_error_); p-Value; Partial η^2^	Repeated Measures ANOVA F(df_time_, df_error_); p-Value; Partial η^2^	Post-Hoc Tests
SDQ subscales							
Emotional problem scale {0–10}	Boys (*n* = 16)	5.69 (2.39) {2–9}	4.44 (1.82) {0–7}	4.56 (2.45) {1–10}	F(2,46) = 1.851; *p* = 0.169; η_p_^2^ = 0.07	n.s.	
	Girls (*n* = 9)	5.89 (2.80) {2–10}	2.78 (1.48) {1–6}	3.56 (2.65) {1–7}		F(2,16) = 7.840; *p* = 0.004; η_p_^2^ = 0.50	Þ
	Overall	5.76 (2.49) {2–10}	3.84 (1.86) {0–7}	4.2 (2.51) {1–10}	*---*	F(2,48) = 9.063; *p* < 0.001; η_p_^2^ = 0.27	Þ¢
Conduct problem scale {0–10}	Boys (*n* = 16)	2.63 (1.82) {0–5}	2.44 (1.31) {0–5}	1.81 (1.22) {0–4}	F(2,46) = 1.327; *p* = 0.275; η_p_^2^ = 0.06	F(2,30) = 4.952; *p* = 0.014; η_p_^2^ = 0.25	Φ¢
	Girls (*n* = 9)	2.00 (1.0) {1–3}	1.00 (1.5) {0–4}	0.56 (0.88) {0–2}		F(2,16) = 4.189; *p* = 0.034; η_p_^2^ = 0.34	¢
	Overall	2.40 (1.58) {0–5}	1.92 (1.53) {0–5}	1.36 (1.25) {0–5}	*---*	F(2,48) = 8.455; *p* = 0.001; η_p_^2^ = 0.26	Φ¢
Hyperactivity scale {0–10}	Boys (*n* = 16)	4.81 (2.90) {0–10}	3.56 (1.90) {0–7}	2.75 (1.98) {1–7}	F(2,46) = 1.408; *p* = 0.255; η_p_^2^ = 0.06	F(1.319,19.785) = 5.707; *p* = 0.02; η_p_^2^ = 0.28;	¢
	Girls (*n* = 9)	2.44 (2.19) {0–6}	2.44 (2.70) {0–8}	1.89 (1.90) {0–5}		n.s.	
	Overall	3.96 (2.86) {0–10}	3.28 (2.30) {0–8}	2.44 (1.95) {0–7}	---	F(1.674,40.167) = 5.346; *p* = 0.012; η_p_^2^ = 0.18;	¢
Peer problem scale {0–10}	Boys (*n* = 16)	2.31 (2.24) {0–8}	2 (2.31) {0–8}	2.06 (1.53) {0–6}	F(2,46) = 0.269; *p* = 0.765; η_p_^2^ = 0.01	n.s.	
	Girls (*n* = 9)	1.67 (1.66) {0–4}	1.67 (1.32) {0–4}	1.89 (0.93) {1–4}		n.s.	
	Overall	2.08 (2.04) {0–8}	1.88 (1.99) {0–8}	2 (1.32) {0–6}	---	n.s.	
Prosocial scale {{10–0}	Boys (*n* = 16)	8.38 (1.36) {{5–10}	8.38 (1.36) {6–10}	8.94 (1.34) {6–10}	F(2,46) = 0.991; *p* = 0.379; η_p_^2^ = 0.04	n.s	
	Girls (*n* = 9)	9.44 (0.88) {8–10}	8.78 (1.79) {5–10}	9.22 (1.56) {6–10}		n.s	
	Overall	8.76 (1.30) {5–10}	8.52 (1.50) {5–10}	9.04 (1.40) {6–10}	---	n.s.	
Total Difficulty Scale {0–40}	Boys (*n* = 16)	15.44 (7.26) {4–29}	12.44 (5.76) {4–26}	11.19 (5.02) {3–21}	F(2,46) = 0.243; *p* = 0.786; η_p_^2^ = 0.01	F(1.482,22.227) = 6.349; *p* = 0.011; η_p_^2^ = 0.30;	¢
	Girls (*n* = 9)	12.00 (5.32) {5–23}	7.89 (5.09) {3–19}	7.89 (4.76) {3–16}		F(2,16) = 5.340; *p* = 0.017; η_p_^2^ = 0.40	¢
	Overall	14.2 (6.72) {4–29}	10.8 (5.86) {3–26}	10.0 (5.09) {3–21}	---	F(2,48) = 11.521; *p* < 0.001; η_p_^2^ = 0.32	Þ¢
**(Very) High (17+) on Total Difficulty Scale (*n* = 9) at Baseline**		**Pre-Test** **Mean (SD)** **{Min-Max}**	**Post-Test** **Mean (SD)** **{Min-Max}**	**Follow-Up** **Mean (SD)** **{Min-Max}**	**Repeated measures ANOVA** **F(df_time_, df_error_); p-value; partial** **η^2^**		**Post-Hoc Tests**
Emotional problem scale		7.22 (2.33) {3–10}	5.56 (1.13) {4–7}	5.00 (1.73) {2–7}	F(2,16) = 7.065; *p* = 0.006; η_p_^2^ = 0.47		¢
Conduct problem scale		3.78 (1.30) {1–5}	3.11 (1.36) {1–5}	2.44 (1.01) {1–4}	F(2,16) = 6.857; *p* = 0.007; η_p_^2^ = 0.46		¢
Hyperactivity scale		6.78 (1.64) {4–10}	4.78 (2.11) {2–8}	3.44 (2.19) {1–7}	F(1.238,9.907) = 7.696; *p* = 0.005; η_p_^2^ = 0.49;		Φ¢
Peer problem scale		4.00 (1.73) {2–8}	2.89 (2.47) {0–8}	2.67 (1.58) {1–6}	F(2,16) = 5.734; *p* = 0.013; η_p_^2^ = 0.42		¢
Prosocial scale		7.89 (1.36) {5–10}	8.11 (1.76) {5–10}	8.56 (1.42) {7–10}	n.s.		
Total Difficulty Scale		21.78 (3.15) {18–29}	16.33 (5.39) {11–26}	13.56 (5.25) {7–21}	F(1.223,9.784) = 19.923; *p* < 0.001; η_p_^2^ = 0.71;		ÞΦ¢

Results for repeated measure ANOVAs and post-hoc tests only shown if significant (*p* < 0.05); n.s. not significant; SD: standard deviation; Þ significant difference between t1 and t2, Φ significant difference between t2 and t3, ¢ significant difference between t1 and t3.

**Table 2 ijerph-18-04530-t002:** Qualitative data interview, participants’ age and gender.

**Caregiver Number**	**Gender**	**Age**
1	female	32
2	female	33
3	female	26
4	female	28
5	male	39
6	male	43
7	female	26
8	female	32
**Facilitator number**		
1	male	25
2	female	28
3	male	32
4	female	25
5	female	39
6	male	24

## Data Availability

The data presented in this study are available on request from the corresponding author. The data are not publicly available due to restrictions of privacy.

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
