# Peer review of "Assessing the Feasibility of Providing a Family Skills Intervention, “Strong Families”, for Refugee Families Residing in Reception Centers in Serbia†"

_ijerph, 2021, doi:10.3390/ijerph18094530_

Round 1
Reviewer 1 Report
Manuscript ID: ijerph-1177596
Title: Assessing the Feasibility of Providing “Strong Families” as a Family Skills Intervention for Refugee Families Residing in Reception Centers in Serbia
Journal: International Journal of Environmental Research and Public Health (IJERPH)
Thank you for the opportunity to review the manuscript “Assessing the Feasibility of Providing “Strong Families” as a Family Skills Intervention for Refugee Families Residing in Reception Centers in Serbia”. This manuscript addresses a relevant topic, as it evaluates the effects of an intervention aimed to improve parenting in an adverse scenario. Paying attention to family relationships is a fundamental step towards promoting mental health. Although the sample is small, I believe it is sufficient to discuss the possible impact that this type of intervention can have on refugee reception centers, or even on other scenarios that serve vulnerable families. However, this manuscript presents a set of flaws that must be addressed before it is processed further.
Below are a few noteworthy problems:
- First, I strongly suggest that the manuscript undergo English editing. It requires editing in terms of structure and style of writing. In addition, there are words in British spelling and others in American spelling (e.g., centres or centers; prioritizing or prioritising; learnt or learned).
- I suggest the inclusion of "Strong Families" among the keywords.
- Avoid using statements like the one that appears on lines 53-54 "... in the world than at any other time in history".
- Avoid using expressions such as very, some, mostly. The statements must present objective data.
- The introduction should be more objective. For example, when the authors mention that there are studies on parenting interventions, they could summarize the findings of those 4 studies in one or two sentences. The authors need to explain more clearly, but succinctly, what is "Strong Families" (is it training? a type of therapy? or counseling?).
- Method section:
- the first sentence is not necessary, as the justification for the study must be (and already is) in the introduction;
- The number of participants must appear in point 2.1, as well as whether they are facilitators, parents, children, etc. and how many of each group participated in the study. The data provided at the beginning of the results session (25 families ... 20 mothers and 5 fathers ... from Afghanistan) should be in the section that describes the participants;
- As section 2.2 describes sample selection, I suggest joining sub-sections 2.1 and 2.2 and naming it 1 Participants and Procedures. The study design can be described as an introductory sentence in the method section, before point 2.1;
- Figure 1 is unnecessary. I believe that it does not even need to be included in the supplementary materials. As mentioned above, the number of participants who completed the intervention must be described in sub-section 2.1;
- Subsection 3 Confidentiality and ethical considerations may be more succinct. Maybe, it may be the final paragraph of subsection 2.1;
- Sub-section Data collection is well described. However, they can include the term "repeated measures" in the second paragraph and explain what is "satisfactory construct and predictive validity" (line 230).
- Programme intervention is objective. I suggest the inclusion of supplementary material that explains what tools are taught, how families practice positive communication, what are the stress relief techniques, what activities are offered to children and what family values are taught.
- Results section:
- Include % and means in the text and avoid using expressions like "mostly some ...";
- Table 1 should be as supplementary material, which is possible if the most important data (% and means that characterize the sample) are described in the text;
- Sub-section 3.1.2: include the p-values when presenting the outcomes;
- Table 2: check formatting (e.g., p must be in italics);
- Figure 2: It is confusing to understand why the n varies from 8 to 11 if there are 22 parents who are in a relationship. Did this analysis consider only those families that scored above the 70th percentile in each PAFAS subscales? If so, the n (or variation of n) must be mentioned in the text;
- Sub-section 3.2: The authors identify the themes as Theme 1, Theme 2 and that's fine. However, subtitles 1.1; 1.2; 2.1; etc. disturb the formatting of the manuscript. I suggest: 2.1 Theme 1: ..... and then the use of bullet points to present the subcategories within the themes.
- In the discussion section, the authors quantify the stress level of the settings where the intervention was applied. It is an unnecessary and controversial statement.
- Pay attention to redundant expressions like "positive improvements" and "punitive punishment".
- In the line 675, the authors mention that the sample consisted of "low ethnic diversity, being entirely Afghan Dari speaking refugees". Ethnic: "relating to or characteristic of a large group of people who have the same national, racial, or cultural origins, and who usually speak the same language" (Cambridge Dictionary).
In conclusion, a major revision is needed before this manuscript is processed further.
I hope these comments are a useful guide for authors to improve the manuscript.
Sincerely,

Author Response
Response in attached document

Reviewer 2 Report
Review of the paper: Assessing the Feasibility of Providing “Strong Families” as a Family Skills Intervention for Refugee Families Residing in Reception Centres in Serbia
Dear authors,
I have found the topic of your quite interesting.
The title is somewhat confusing. Please revise according to its meaning.
The template was not correctly followed. Please see IJERPH authors guidelines.
The study is well conducted but the presentation is poor. I believe that is necessary to present it nicely, by improving tables quality.
Also, in 3.1.3 there is some grey that needs to be removed.
In 3.2 Qualitative results it would be nice to have organized differently:
First, a table identifying the participants with sex, age and others. This is important to associate with their comments and views
Second, another table where the main thematic dimensions identified and the corresponding quotes. This is a more reflexive approach that shows authors efforts to analyse discourses content.
Good luck!
Author Response
Response in additional attached file

Round 2
Reviewer 1 Report
Manuscript ID: ijerph-1177596
Title: Assessing the Feasibility of Providing “Strong Families” as a Family Skills Intervention for Refugee Families Residing in Reception Centers in Serbia
Journal: International Journal of Environmental Research and Public Health (IJERPH)
The authors did an excellent job of improving the manuscript. However, the manuscript still has presented minor flaws:
I still think the introduction could be more objective. As I said, "when the authors mention that there are studies on parenting interventions, they could summarize the results of those 4 studies in one or two sentences". The authors added a conclusion about the studies. However, my suggestion was precisely to shorten that paragraph.
The statement "Research that provides insights in understanding why, how and in which settings psycho-social programmes can effectively operate in humanitarian contexts are greatly lacking [27]" is not necessary in section 2, as it is a justification for the study. The authors already justify the study in the introduction.
The authors stated that 26 families were invited to participate in the study, with one family leaving the camp before the program started and, therefore, the total was 25 families. However, then they mention "One other family took part in the pre-intervention assessment, but then moved out of the Reception Centre before intervention and hence was excluded". This statement is confusing, as they said there were 25 families. Would there be 27 initially?
Section 2.3 must be 2.2; as well as 2.4 is 2.3.
In the statement “Mothers participated with 9 daughters and 11 sons, 55% and 45% respectively, fathers however only with sons" is the order of the percentages not reversed?
I had just made a suggestion on how to order subsection 3.2. However, I understand and accept the maintenance of numbering the themes.
Kind regards,

Author Response
We thank reviewer 1 again for taking time to check our responses and make further useful comments. Our responses follow:
I still think the introduction could be more objective. As I said, "when the authors mention that there are studies on parenting interventions, they could summarize the results of those 4 studies in one or two sentences". The authors added a conclusion about the studies. However, my suggestion was precisely to shorten that paragraph.
We have removed the final paragraph that outlines the four studies as well as the additional concluding sentence. In place, we have added the following sentence:
‘However, there are four recent exceptions of studies indicating the effectiveness of parenting and family skills interventions, in refugee and displacement contexts, in reducing child maltreatment and improving parental and child mental health in humanitarian settings [10,19,20,21]’.
The statement "Research that provides insights in understanding why, how and in which settings psycho-social programmes can effectively operate in humanitarian contexts are greatly lacking [27]" is not necessary in section 2, as it is a justification for the study. The authors already justify the study in the introduction.
This has been removed and thus the reference numbering has been edited to reflect the removal of number 27.
The authors stated that 26 families were invited to participate in the study, with one family leaving the camp before the program started and, therefore, the total was 25 families. However, then they mention "One other family took part in the pre-intervention assessment, but then moved out of the Reception Centre before intervention and hence was excluded". This statement is confusing, as they said there were 25 families. Would there be 27 initially?
We have clarified this paragraph in text as follows:
‘Overall, 26 families were invited to participate in the study, 12 from Vranje reception Centre, 11 from Preševo reception Centre and 3 from Bujanovac reception Centre. All families agreed to participate, met the inclusion criteria and participated in the programme. One family was excluded after leaving the camp after taking part in the pre-intervention assessment, therefore, overall 25 families participated in the study’.
Section 2.3 must be 2.2; as well as 2.4 is 2.3.
Thank you for noting this, the numbering in section 2 has been amended.
In the statement “Mothers participated with 9 daughters and 11 sons, 55% and 45% respectively, fathers however only with sons" is the order of the percentages not reversed?
The reviewer is correct, and we thank them for pointing this out. The percentages should be reverted, it has been edited to
‘Mothers participated with 9 daughters and 11 sons, 45% and 55% respectively, fathers however only with sons’.
I had just made a suggestion on how to order subsection 3.2. However, I understand and accept the maintenance of numbering the themes.
Thank you for accepting this.

Reviewer 2 Report
Dear authors,
Thanks for the revisions made.
Regards,
Author Response
Thank you for accepting our revision